# Estimation of supply and demand for public health nurses in Japan: A stock-flow approach

Kazuya Taira[1]*, Masanao Horikawa[2], Takahiro Itaya[3], Rikuya Hosokawa[1], Misa Shiomi[1]

1 Human Health Sciences, Graduate School of Medicine, Kyoto University, Kyoto, Japan, 2 Human Health Sciences, Faculty of Medicine, Kyoto University, Kyoto, Japan, 3 Department of Healthcare Epidemiology, Graduate School of Medicine and Public Health, Kyoto University, Kyoto, Japan

* taira.kazuya.5m@kyoto-u.ac.jp

## Abstract

Public health nurses (PHNs) play a central role in community health in Japan, and the number of certified PHNs has decreased since 2011. However, thus far, no study has estimated the supply and demand of PHNs in the coming years. The present study aimed to estimate the future balance between supply and demand for PHNs in Japan. This simulation study adopted a stock-flow approach using data from a survey of local governments, log data from a recruitment information website, and government statistics. The supply was estimated by adding up the numbers of newly hired PHNs and job changers to that of PHNs from the human resource pool. The demand was estimated from the number of PHNs needed in the future calculated by demographics assumed to affect the demand for PHNs. At the current job change rate to PHNs, the gaps between future supply and demand of PHNs were expected to be –494 to –50 in 2025, –1,007 to –435 in 2030, and –772 to –330 in 2035. If the job change rate would be 1.5 times, the gaps were estimated to be 285 to 729 in 2025, –431 to 141 in 2030, and –182 to 260 in 2035. If the job change rate would be 2.0 times, the supply was sufficient for all estimated years. The number of PHNs working in Japanese local governments is estimated to decline until 2035, resulting in a shortage. Policy makers should consider early measures to adjust the future number of PHNs.

## Introduction

Public health centers play many roles in maintaining and promoting community health, such as maternal and child health, enhancement of residents' health, preventing the spread of infectious diseases, and medical care and social support for patients with intractable or rare diseases [1]. Since the coronavirus pandemic of 2020, public health centers have taken on much responsibility for coronavirus disease 2019 (COVID-19) control, including the identification of infected individuals. Consequently, the workload of public health nurse (PHNs) and other public health center staff has increased rapidly.

**Data availability statement:** Data used in this study are not available for public access. Participant consent for sharing survey results was not obtained, and redistribution permission for data from the "Komuin" website is not granted. The authors obtained data from the "Komuin" website by contacting the site administrator via the inquiry form (https://comin.tank.jp/mail.php). As no direct contact details are provided for the site administrator, this inquiry form serves as the only contact method. The authors do not have any special data access privileges. Additionally, raw government statistical data are not held by the authors but can be accessed through aggregation cited in the references of this paper. For further inquiries about government statistical data, please contact the government statistics portal, e-Stat, through their inquiry form (https://www.e-stat.go.jp/contact) or directly reach out to the relevant Japanese government ministries cited in the references.

**Funding:** This work was supported by FY 2021 Kusunoki 125 of Kyoto University 125th Anniversary Fund to MS and the Japan Society for the Promotion of Science (JSPS KAKENHI; grant number: 22K17549 to KT). The funders had no role in study design, data collection and analysis, decision to publish, or preparation of the manuscript. there was no additional external funding received for this study.

**Competing interests:** The authors have declared that no competing interests exist.

Although PHNs can work in several types of workplaces, such as companies and hospitals, most work in public health centers owned by the local government [2]. PHNs in Japan are nursing professionals licensed by the Ministry of Health, Labour, and Welfare to engage in health guidance [3]. In Japan, to obtain a license as a PHN, one must hold a nursing license. Therefore, the human resource pools of nurses and PHNs overlap, and the promotion of nurses' employment also affects the recruitment of PHNs. In addition, the older adult population in Japan is expected to continue to grow [4], increasing the social need for medical and nursing care, and resulting in the requirement for more nurses and PHNs [5]. PHNs have difficulty hiring foreigners such as nurses and caregivers because other countries do not have PHN positions that perform the same duties as in Japan or the appropriate national qualifications, and they lack a system to help foreigners obtain PHN qualification [6].

Under the Local Autonomy Law, local governments have the right to determine the number of public health personnel they employ, which has been declining each year [7]. This downward trend is attributed to Japan's declining population; however, the number of PHNs continues to increase due to the increasing needs for nursing care and welfare services for the aging population as well as disaster preparedness.

## Changes in the training system and supply of qualified PHNs

Until 2011, PHN education was undertaken by all undergraduate nursing students. In response to changes in the environment of local residents and the growing complexity of health issues as society matures, the PHN training system was changed to allow for a selective course in university undergraduate programs to train highly specialized professionals.

As a result of the revision in the PHN training system [8,9], the number of PHN training schools saw a significant decline from 15,694 in 2011 to 8,472 in 2012. This shift occurred due to the transition from a system where all undergraduate nursing students were eligible for certification to a capacity-based elective system or graduate-level education.

PHNs play an important role in natural disaster response and infectious disease control, including COVID-19, and their demand is expected to increase in the future [10]. However, estimates of the future demand for and supply of PHNs, and changes in the demand for PHNs since the COVID-19 pandemic have not yet been reported.

## Estimation of supply and demand in the medical profession

There are several reports on future estimates of the number of medical doctors and registered nurses [11,12]. In the United States, the number of registered nurses is highly skewed by state, and it is estimated that some states will have an excess of nurses while others will have a shortage by 2030 [13]. In Japan, the Ministry of Health, Labour, and Welfare has estimated that approximately two million nurses are needed to meet the future demand for medical care in terms of the number of patients by 2025 [14]. However, the number of employed nurses in 2019 was 1.58 million, reflecting a flat trend since 2016 [2], and the number of nurses may be insufficient by 2025.

Although the demand for nurses can be estimated from the number of patients and beds [15], PHNs provide support not only to the patients but also to healthy residents, and their work is as diverse as the roles of public health centers [1], making it difficult to estimate the demand based on workload. Therefore, this study estimated the demand for PHNs using demographic data of residents who are likely to be targeted for support by PHNs.

This study aimed to investigate the increase in the demand for PHNs before and after the COVID-19 pandemic and estimated the future supply and demand of PHNs.

## Methods

### Data source

This was a simulation study based on a questionnaire survey of Japanese local governments, secondary data from government statistics, and log data from a website [16–21]. The ethical review of this study was waived by the ethics committee of the institution with which the authors are affiliated because it was a questionnaire survey that did not involve personal information and secondary use of published data.

### Questionnaire survey of local governments in Japan

A questionnaire survey, sent by mail between November and December 2021, was conducted involving all Japanese local governments that have public health centers, which are called "Hokenjo" in Japanese [22]. We identified from the target local governments' websites the department in charge of human resource management, such as recruitment, leave of absence, retirement, and staffing, and asked the department preparing the responses for questionnaire. In the Japanese government system, the responses were generally double-checked by the staff and manager in the department and the answer was given in the name of the head of the department. The targeted local governments included those of 47 prefectures, 20 designated cities, 54 core cities, 23 special wards, and 6 other government ordinance cities. In Japan, the roles of public health centers in prefectures and municipalities are different, and the 150 local governments targeted in this study were those responsible for dealing with patients with COVID-19. Local governments where public health centers are located consist of prefectures that unite cities, towns, and villages, and cities with large populations, such as ordinance-designated cities with populations of 500,000 or more and core cities with populations of 200,000 or more. Including other smaller cities, towns and villages, there are 1,718 local governments in Japan. However, only local government with public health centers were included in this survey because large local governments, which have more discretion over personnel costs, are more sensitive to increasing or decreasing the number of PHNs, and public health centers are also likely to be significantly affected by COVID-19 pandemic. This survey aimed to clarify the employment status of PHNs in public health centers owned by local governments for the last five years (2017–2021); the survey items included the number of full-time PHNs hired, number of new graduates among the full-time PHNs hired, number of part-time PHNs hired, and number of full-time PHNs who quit or retired (S4 Fig). However, since the objective of this study was to determine the number of staff engaged in public health nursing services, we requested that PHNs and RNs working in hospitals and clinics operated by local governments be excluded from the count.

### Log data public employee recruitment information website

Log data for recruitment information, which were posted on the public employee recruitment information website for civil servants and PHNs working in local governments, named "Komuin," [16] were taken from April 2013 to July 2021. The variables provided by the owner of the website included the name of the hiring local government, date of the hiring test, and age requirement.

### Government statistics

The number of successful applicants for the national examination for PHNs was obtained from the Ministry of Health, Labour, and Welfare's press release [17]. The number of PHNs employed and their ages were obtained from the Report on Public Health Administration

and Services [18]. The future estimates of demographic statistics used to calculate the demand for PHNs were taken from the following sources: the number of births estimated by the National Institute of Social Security and Population Studies [19]; the number of people certified as requiring long-term care estimated by the Ministry of Economy, Trade, and Industry [20]; and the number of patients with cancer estimated by the National Cancer Center [21].

## Analyses

The supply and demand of PHNs were estimated using a stock-flow approach. The supply estimate of PHNs was calculated based on the following equations using the descriptive statistics of the data:

$Supply\ Estimates\ at\ Ti$

$= The\ number\ of\ newly\ graduated\ public\ health\ nurses\ at\ Ti$

$+ The\ number\ of\ people\ transitioning\ to\ public\ health\ nurses\ at\ Ti$

$The\ number\ of\ people\ transitioning\ to\ public\ health\ nurses\ at\ Ti$

$= The\ potential\ public\ health\ nurse\ resource\ pool\ at\ Ti$

$\times Percentage\ of\ job\ changers\ entering\ public\ health\ nursing$

$The\ potential\ public\ health\ nurse\ resource\ pool\ at\ Ti$

$= The\ number\ of\ nurses\ within\ the\ employable\ age\ range\ at\ Ti$

$- The\ number\ of\ nurses\ already\ working\ as\ public\ health\ nurses$

Each parameter was estimated as follows.

## Supply-related parameter 1. Estimates of the number of newly graduated PHNs

The number of newly graduated PHNs was calculated based on the statistics of those who passed the national examination for obtaining a PHN license [17]. We assumed that the average number of PHNs would continue to be 7,281 per year, which was the average number from 2016 to 2020, when the number of newly licensed PHNs in the country declined due to the change in the training system. The percentage of new graduates from training schools who become PHNs was reported to range from 6% to 12%, and this study assumed it to be 10% [23,24].

## Supply-related parameter 2. Estimates of the number of PHNs within the age limit for employment

The age limit was set at 37 years, which is the average upper age limit for PHN recruitment examinations, calculated from the log data in recruitment information [16]. In Japan, the typical age for nurses who graduate from a four-year university and obtain a new PHN license is 22 years, excluding those who stay in school, are wanderers, or transfer with a bachelor's degree. Therefore, using the same data as in parameter 1 [17], we assumed that the age of the national exam licensees was 22 years, and the age of the licensees within 15 years, of the time point we aimed to evaluate was 37 years or less.

## Supply-related parameter 3. Estimates of the number of nurses already working as PHNs

The number of employed PHNs within a five-year age group in 2020 was obtained from government statistics to estimate the number of nurses already working as PHNs [18]. The age at which PHNs passed the national exam was set at 22 years, and we calculated backward to see what year the PHNs working in each age category obtained their licenses. The number of nurses already working as PHNs in the five-year age group was divided by 5, and the average was the number of nurses already working as PHNs for each year in the five-year age group.

## Supply-related parameter 4. Estimates of the percentage of job changes by PHNs

The number of PHNs hired in 2021 was 2,447[25], and excluding the estimated number of new graduates PHNs, that is 728, 1,719 had changed jobs from the human resource pool to PHNs. The number of people in the talent pool was 141,597, and the number of people changing jobs was 1,719, representing 1.2% of the total.

The demand estimate of PHNs was calculated based on the following equations using demographic data and linear regression equation.

$$\begin{aligned} \text{Demand Estimates at Ti} \\ = \text{The number of additional public health nurses needed at Ti} \\ + \text{The number of new public health nurses hired by local governments in 2020} \end{aligned}$$

$$\begin{aligned} \text{The number of additional public health nurses needed at Ti} \\ = \text{The projected number of public health nurses needed at Ti} \\ - \text{The number of public health nurses employed by local government in 2020} \end{aligned}$$

$$\begin{aligned} \text{The projected number of public health nurses needed at Ti} \\ = \left[ \text{Regression coefficient} \times \text{Demographic parameters at Ti} \right] + \text{Intercept} \end{aligned}$$

## Demand-related parameter 1. Demographic parameters

The number of PHNs that will be needed in the future was calculated from the demographic parameters. According to the Survey of Public Health Nurses' Activities (Area Survey) [25], a survey on the workload of PHNs, there are nine fields of work for PHNs: maternal and child, mental health, elderly, incurable diseases, child welfare, occupational health, infectious diseases, disabilities, and health promotion. Three specialists (authors: KT, RH, MS) with at least three years of experience as PHNs and education at schools that train PHNs discussed the nine fields that have a strong impact on the demand for the number of PHNs and narrowed the list to three fields: maternal and child health, older adults, and health promotion. The demographic statistics used to project the number of PHNs in each field were the number of births [19] in the maternal and child field, the number of people certified as needing long-term care [20] in the older adult field, and the number of patients with cancer [21] in the health promotion field.

## Demand-related parameter 2. Estimates of regression coefficient and intercept

A linear regression analysis was performed with the number of PHNs as the dependent variable and each demographic variable as the independent variable to estimate the regression

coefficient [B] and intercept. An adjusted coefficient of determination [adjusted $R^2$] was also calculated to check the explanatory power of the model. The number of PHNs needed in the future was calculated by substituting the demographic variable at the time point Ti we wanted to estimate.

### Demand-related parameter 3. Number of PHNs working in local governments in 2020 and new PHNs hired by local governments in 2020

Statistics from the Report on Public Health Administration and Services were used [19]. Using the number of new PHNs hired by local governments in 2020 as the baseline for demand, the demand for PHNs was defined as the difference between the estimated number of PHNs needed in the future and the number of PHNs in employment as of 2020 (the current demand). Since only five-year estimates could be calculated, we divided the total demand for five years by five and treated the average as the estimate for each year.

Furthermore, crises such as major natural disasters (e.g., Kumamoto earthquake in 2016, Tohoku earthquake in 2011) and emerging infectious diseases (e.g., Bird-flu: H7N9 in 2013, Middle East Respiratory Syndrome in 2012, Severe Acute Respiratory Syndrome in 2002) occur in Japan every few years. Along with those crises, the government has set aside budgets to increase the supply of PHNs. Further to battle the COVID-19 pandemic, it has also prepared a budget to increase prefectural PHNs by 1.5 times [26]. In the near future, large-scale natural disasters are expected to occur in the disaster management plan, including a Nankai Trough earthquake, directly under the Tokyo metropolitan area, and the eruption of Mt. Fuji [27]. Therefore, the study also examined a situation in which 1.5 and 2 times as many PHNs would be needed, assuming a crisis of the same magnitude as the COVID-19 pandemic.

The ethical review of this study was waived by the Kyoto University Graduate School and Faculty of Medicine Kyoto University Hospital Ethics Committee because it was a questionnaire survey that did not involve personal information and secondary use of published data.

## Results

### Questionnaire survey of local governments in Japan

The response rate of the survey was 54.0% (81/150). For each variable, we compared the data for 2019 and 2020, that is, before and after the COVID-19 pandemic, respectively. The mean number of full-time PHNs for local governments was 103.2 (Standard Deviation: SD = 75.9) in 2019 and 104.5 (SD = 76.4) in 2020. The mean number of part-time PHNs for local government was 11.7 (SD = 14.9) in 2019 and 22.4 (SD = 53.6) in 2020. The mean number of PHNs who quit or retired for local government was 5.0 (SD = 4.3) in 2019 and 4.9 (SD = 4.5) in 2020. The number of PHN recruitment examinations for local government was 1.3 (SD = 1.0) in 2019 and 1.6 (SD = 1.1) in 2020 (Table 1).

### Log data from the public employee recruitment information website "Komuin"

The mean number of recruitment information posted on "Komuin" from 2013 to 2020 was 1,009 (SD = 125). Japan has 47 prefectures and 1,718 municipalities for 1,765 local governments, and the coverage rate was roughly 57.2% (1,009/1,765). PHN recruitment information by local governments showed a similar bimodal pattern every year (S1 Fig). The average age limit for the employment of PHNs calculated from the recruitment information was 34.1 (SD = 5.7) years in 2013, 36.2 (SD = 6.7) years in 2017, and 37.5 (SD = 7.2) years in 2021, reflecting an upward trend.

Table 1. Summary of PHNs employment status statistics for local governments.

| | 2017 | 2018 | 2019 | 2020 | 2021 |
|---|---|---|---|---|---|
| **Number of full-time public health nurses (PHNs) for local governments** | | | | | |
| Number of local governments | 81 | 81 | 81 | 81 | 81 |
| Mean (Standard Deviation) | 99.7 (74.5) | 101.4 (74.0) | 103.2 (75.9) | 104.5 (76.4) | 108.4 (79.3) |
| Median | 79 | 84 | 86 | 86 | 88 |
| (Interquartile range) | (58 – 119) | (60 – 119.5) | (61 – 120) | (63.5 – 119.5) | (63.5 – 122.5) |
| (Minimum – Maximum) | (21 – 524) | (22 – 526) | (22 – 547) | (24 – 554) | (22 – 570) |
| **Number of part-time PHNs for local governments** | | | | | |
| Number of local governments | 68 | 68 | 68 | 72 | 70 |
| Mean (Standard Deviation) | 12.0 (15.4) | 11.9 (15.8) | 11.7 (14.9) | 22.4 (53.6) | 18.8 (33.5) |
| Median | 6 | 6 | 6 | 10 | 11 |
| (Interquartile range) | (2 – 17) | (2 – 15) | (1.25 – 17.75) | (3 – 19.75) | (3 – 20.25) |
| (Minimum – Maximum) | (0 – 89) | (0 – 97) | (0 – 85) | (0 – 405) | (0 – 246) |
| **Number of PHNs who quit or retired for local governments** | | | | | |
| Number of local governments | 76 | 76 | 76 | 76 | 75 |
| Mean (Standard Deviation) | 4.5 (4.0) | 4.3 (3.5) | 5.0 (4.3) | 4.9 (4.5) | 1.1 (1.8) |
| Median | 3 | 3 | 4 | 3.5 | 0 |
| (Interquartile range) | (2 – 6) | (2 – 6) | (2 – 7) | (2 – 6.75) | (0 – 1) |
| (Minimum – Maximum) | (0 – 20) | (0 – 15) | (0 – 25) | (0 – 24) | (0 – 7) |
| **Number of PHN recruitment examinations for local governments** | | | | | |
| Number of local governments | 81 | 81 | 81 | 81 | 81 |
| Mean (Standard Deviation) | 1.3 (1.0) | 1.4 (1.0) | 1.3 (1.0) | 1.6 (1.1) | 1.5 (1.1) |
| Median | 1 | 1 | 1 | 1 | 1 |
| (Interquartile range) | (1 – 1) | (1 – 2) | (1 – 1) | (1 – 2) | (1 – 2) |
| (Minimum – Maximum) | (0 – 5) | (0 – 9) | (0 – 9) | (0 – 9) | (0 – 9) |

## Government statistics

The number of licensed PHNs has decreased by almost half, from approximately 17,000 to 8,000, due to the revision of the training system (Fig 1). The average number of nurses who passed the national exam from 2016 to 2019 after the number of certified PHNs decreased was 7,281 per year.

The number of PHNs working in local governments has been steadily increasing, from 31,581 in 1996 to 55,595 in 2020 (S2 Fig). The number of newly hired PHNs in 2020 was 2,185. The percentage of total PHNs aged 60 and older almost doubled, from 3.6% (n = 1,142) in 1996 to 8.1% (n = 4,518) in 2020.

## Supply estimates of PHNs

Supply estimates are shown in Fig 1. For the number of newly graduated PHNs, it was estimated that 728 would be employed annually after 2021. For the number of people changing their job to PHNs, first, the number of PHNs within the employable age range is estimated to be 157,704 in 2025, 125,733 in 2030, and 116,752 in 2035. Excluding the number of nurses already working as PHNs, the human resource pool of potential PHNs would be 129,772 in 2025, 95,933 in 2030, and 98,295 in 2035. In 2020, approximately 1.2% (n = 1,948) of people

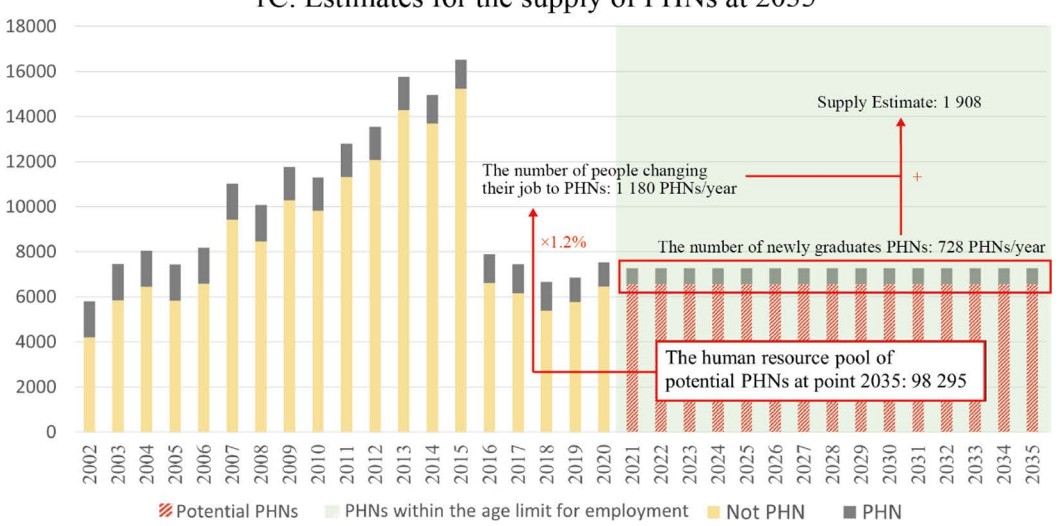

**Fig 1. Changes in the number of public health nurses certified and supply estimates.**

from the human resource pool changed jobs to PHNs, and the estimated number of people changing jobs to PHNs at the same rate would be 1,557 in 2025, 1,151 in 2030, and 1,180 in 2035. The estimated supply of PHNs, including new graduates and job changes, is expected to be 2,285 in 2025, 1,879 in 2030, and 1,908 in 2035.

### Demand estimates of PHNs

The future number of PHNs needed, which is estimated based on the regression equation applied to each bivariate, is 58,567 in 2025, 60,251 in 2030, and 62,260 in 2035 based on the number of births [20]; 58,067 in 2025, 61,574 in 2030, and 64,050 in 2035 based on the number of persons requiring certified long-term care [21]; and 56,347 in 2025, 56,990 in 2030, and 57,257 in 2035 based on the estimated number of patients with cancer [22] (S3 Fig, Fig 2). The averages were calculated and converted to a yearly change in PHNs demand, which is expected to increase by 337–594 persons based on the number of births, 494–701 persons based on a person's need of care, and 53–150 persons based on the number of patients with cancer.

### Comparison of supply and demand of PHNs

A comparison of the supply and demand of PHNs is shown in Fig 3. Calculated at the current job change rate from the current talent pool, the supply of PHNs is expected to be 2,285 in 2025, 1,879 in 2030, and 1,908 in 2035. The demand for PHNs is estimated to be 2,779 in 2025, 2,522 in 2030, and 2,586 in 2035 based on the number of births; 2,679 in 2025, 2,886 in 2030, and 2,680 in 2035 based on the number of persons requiring nursing care; and 2,335 in 2025, 2,314 in 2030, and 2,238 in 2035 based on the number of cancer patients.

Calculating the gap between supply and demand for future PHNs, the supply was insufficient for demand in all estimated years 2025, 2030, and 2035 when the current rate of job change to PHNs was used. If the job change rate to PHNs would be increased by 1.5 times, the supply was sufficient in 2025, but in 2030 and 2035, there was a shortage of up to 431 positions. Furthermore, if the job change rate would be increased to 2 times, the supply was sufficient for all estimated years (Fig 4).

### Discussion

This study estimated the supply and demand of PHNs, who play a central role in Japan's public health policies. Estimates based on the current job change rate to PHN indicated that the demand-supply gap would be insufficient in all estimated years, and even with a 1.5-fold increase in the job change rate, there might be a slight shortage after 2030. Although the number of newly certified PHNs in 2016 (Fig 1) has declined sharply, the impact on the human resource pool of PHNs will not be felt until after 2030 or later as there is a time gap before they reach the upper age limit for recruitment. From Fig 3, it can be inferred that to maintain a balance between the supply and demand of health workers after 2025, the supply needs to be increased by 1.5 or 2.0 times. In terms of the number of PHNs, an increase of 578–779 PHNs would be required. Even at present, PHNs are unevenly distributed throughout the region, with shortages noted in underpopulated areas [28], and the potential for such areas to expand in the future is high if no measure is taken.

In this study, we adjusted for the number of job change rates, but there are three possible ways to maintain the current levels of PHN supply and demand. The first is to increase the number of newly certified PHNs, the second is to set the recruitment age limit to a higher age and expand the human resources pool, and the third is to improve the job change rate from the human resources pool. The number of newly certified PHNs would be required to

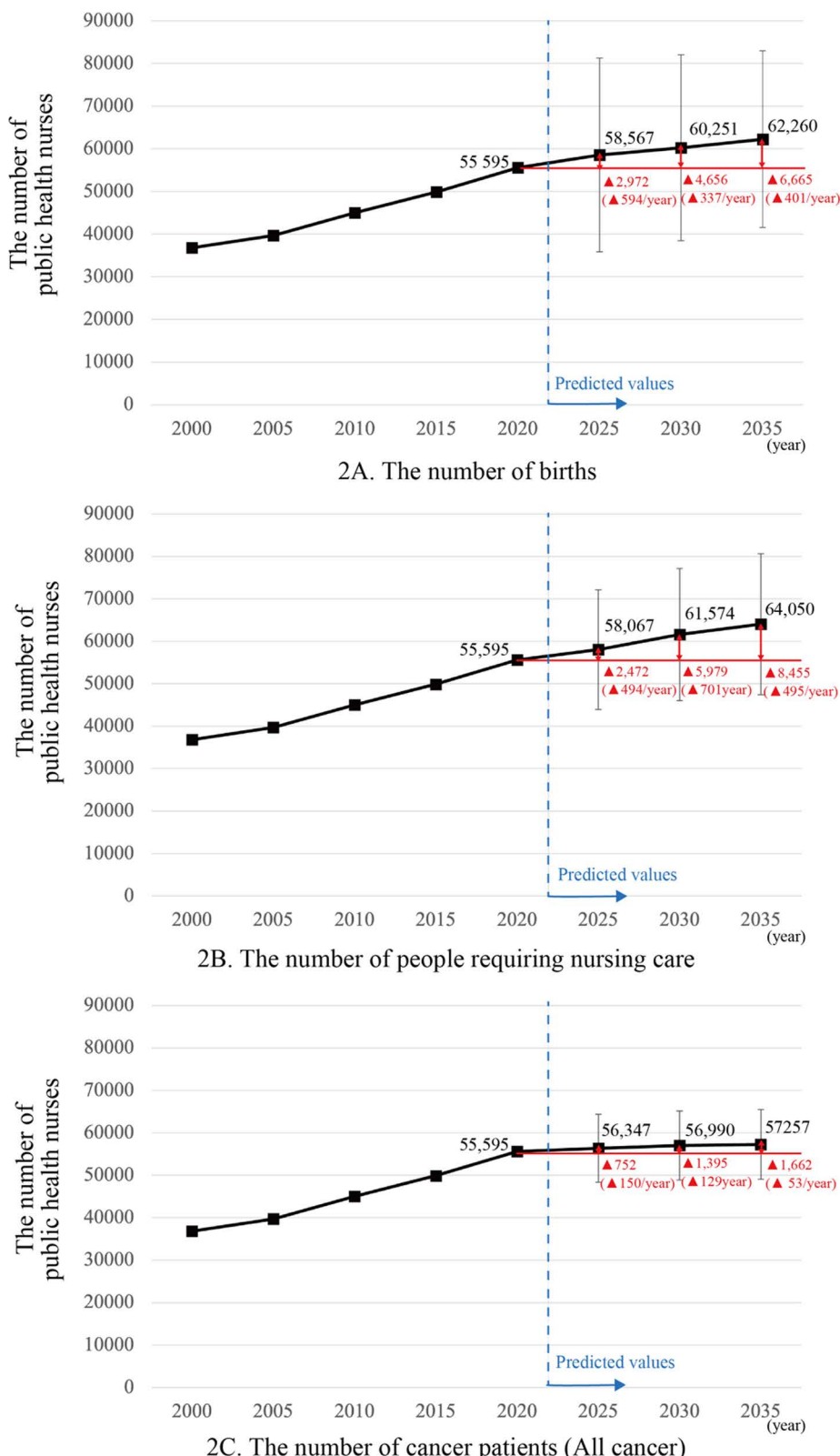

**Fig 2. Estimation of the number of public health nurses needed in the future by demographics.**

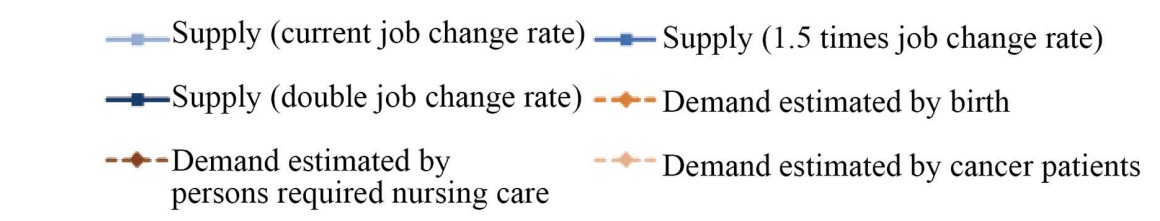

| Supply | 2025 | 2030 | 2035 |
|---|---|---|---|
| Newly graduate public health nurse | 728 | 728 | 728 |
| Job change | | | |
| Current job change rate | 1557 | 1151 | 1180 |
| 1.5 times job change rate | 2336 | 1727 | 1770 |
| Double job change rate | 3114 | 2302 | 2360 |
| Demand | | | |
| Public health nurses newly hired in 2020 | 2185 | 2185 | 2185 |
| Increased demand by the number of | | | |
| birth | 594 | 337 | 401 |
| persons required nursing care | 494 | 701 | 495 |
| cancer patients | 150 | 129 | 53 |

**Fig 3. Comparison of future supply and demand of public health nurses.**

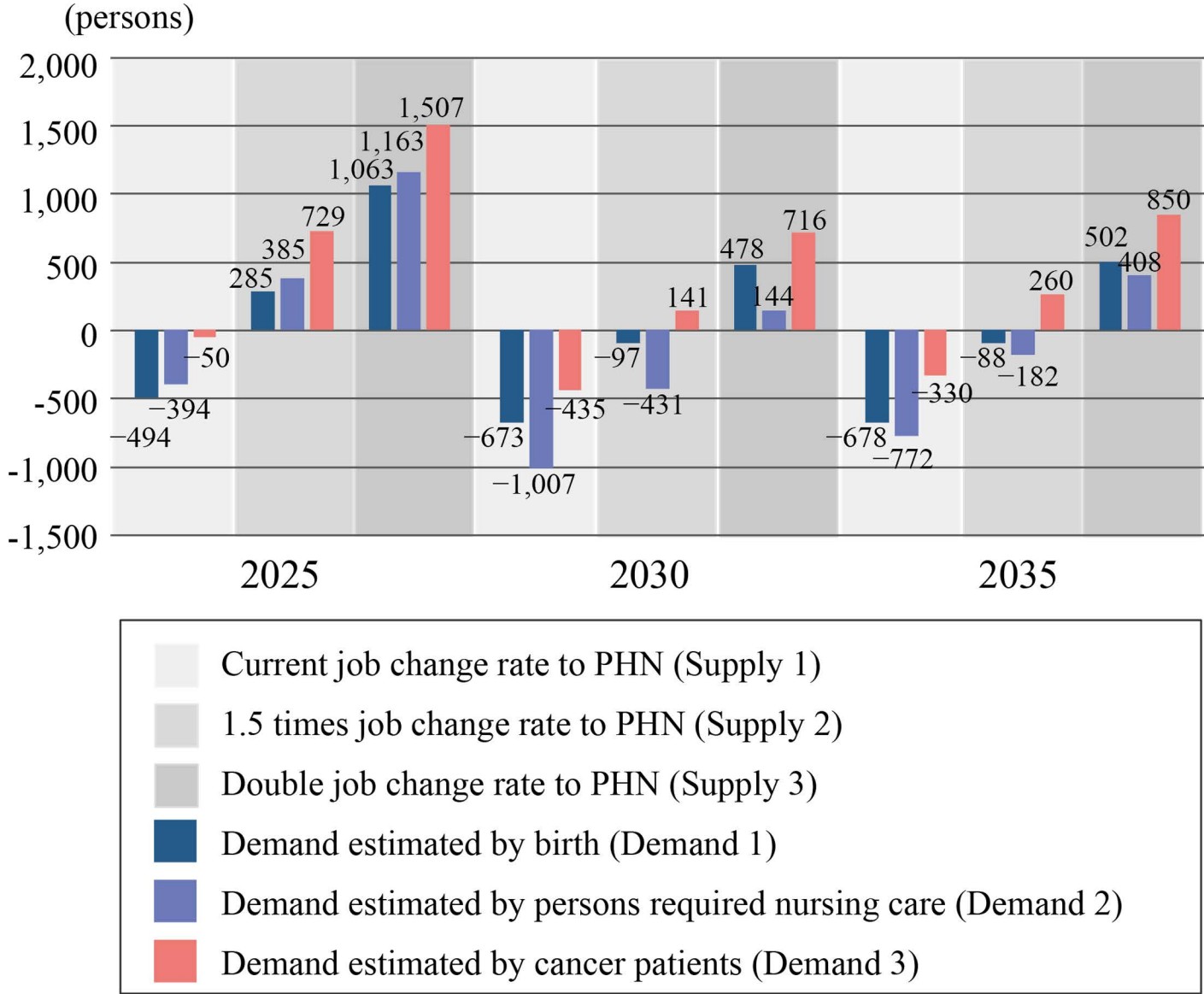

**Fig 4. Gaps of future supply and demand of public health nurses.**

almost double, given the difference between future supply and demand of PHNs. As PHN training programs are strictly regulated by law [29] and capacity issues in training schools and the local governments where they practice, it would be difficult to double the number of newly certified PHNs. Alternatively, to raise the upper age limit for recruitment, it is necessary to raise it enough to make up for the increase in the human resource pool by approximately 50,000–65,000, calculated by the current job change rate. It is equivalent to raising the upper age limit by 5–7 years, that is, bringing the upper age limit to around 42–44 years of age. Although the upper age limit is also on the rise, this may be due to the rapid increase in the number of PHNs aged over 60, possibly due to reappointment positions after retirement. Administrative staff in Japan, including PHNs, has adopted a seniority-based salary system in which their salary increases according to age [30,31]. Therefore, raising the

recruitment age limit for PHNs leads to financial pressure. Furthermore, the seniority-based salary structure is linked to other professional and clerical positions, which may cause friction between job categories. While raising salaries is the established strategy for increasing the job change rate, it would be difficult to take aggressive measures for the same reason. It is also necessary to consider that most of the qualified personnel in the human resource pool are likely to be working as registered nurses in hospitals and clinics, and that, as with PHNs, there is a shortage of human resources [14,28], and thus, a structure of competition for human resources. For these reasons, in reality, these three methods should be combined to take measures.

## Limitations

This study has some limitations. First, as the age groups in the published statistics could not be unified, the number of PHNs aged under 39 years was calculated in the estimation of the supply of PHNs. Second, this study did not consider the demand for part-time PHNs. The results of Table 1 suggest that the mean number of part-time PHNs is increasing, and their actual demand could be much higher. Thirdly, the survey for local governments conducted in this study was conducted for prefectures and large cities and did not take into account the number of PHNs in smaller local governments. However, smaller local governments were excluded from the study because some of them have no choice for increase or decrease in the number of PHNs, with only 1 or 2 PHNs assigned to them in the first place, and they may be affected by other factors, such as the shortage of human resources in rural areas. Fourthly, although simple univariate prediction models were used to estimate demand of PHNs in this study, more accurate model building might be done using multivariate analysis. However, more data needs to be accumulated, and this is an issue to be addressed in the future. Finally, we could not make estimates that consider the uneven distribution of PHNs across regions or complex estimates that combine multiple measures for the estimated shortage of PHNs; however, we aim to address these issues in future studies.

We made estimates that are as conservative as possible for the missing data needed for estimation in this study. As the number of PHNs in a community has been related to the standardized mortality ratio [30] and long-term nursing care utilization rate [32] of the local population, a stable supply of PHNs is an important public health issue in Japan. Since the advent of the COVID-19 pandemic, the Japan Nurses Association [33] has strengthened its public relations efforts to secure PHNs, while the Ministry of Internal Affairs and Communications Autonomous Finance Bureau [26] has secured a budget to increase the number of PHNs working in public health centers by 1.5 times. In the upper trend of the demand for PHNs, including the possibility of a sudden increase due to crisis, measures to prepare for a sharp decline in the human resource pool of PHNs after 2030 should be considered carefully. This study provides an empirical basis for considering the capacity of future PHN training schools and the employment strategy of local governments that hire PHNs. The results are expected to contribute to the stable recruitment of PHNs, which in turn will help realize healthy and sustainable local communities.

## Conclusions

This study aimed to estimate the future supply and demand of PHNs working in Japanese local governments. After 2030, the human resources pool of PHNs will be drastically reduced and there may be a shortage of human resources. As it is time-consuming to secure human resources due to the system for training PHNs and the employment environment, early measures to secure a sufficient number of PHNs are necessary.

## Supporting information

**S1 Fig. Trends in the demand for employment of public health nurses by month from 2013 to 2021.**
(DOCX)

**S2 Fig. Changes in the distribution of public health nurses working in local governments by age group.**
(DOCX)

**S3 Fig. Correlations between demographic variables and the number of public health nurses.**
(DOCX)

**S4 Fig. Survey on the Employment Status of Public Health Nurses (PHNs) during the Corona Disaster Survey Questionnaire.**
(DOCX)

## Acknowledgments

We would like to thank the administrator of the employment information site "Komuin" for providing the log data for this study and Editage (www.editage.com) for English language editing.

## Author contributions

**Conceptualization:** Kazuya Taira.

**Data curation:** Masanao Horikawa.

**Formal analysis:** Kazuya Taira.

**Funding acquisition:** Kazuya TairA, Misa Shiomi.

**Investigation:** Kazuya Taira, Masanao Horikawa, Rikuya Hosokawa, Misa Shiomi.

**Methodology:** Kazuya Taira, Takahiro Itaya.

**Supervision:** Takahiro Itaya, Misa Shiomi.

**Validation:** Masanao Horikawa, Takahiro Itaya.

**Visualization:** Masanao Horikawa.

**Writing – original draft:** Kazuya Taira.

**Writing – review & editing:** Takahiro Itaya.

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
