## [Decision Letter · Decision Letter 0]

19 Dec 2023

PONE-D-23-19276Estimation of supply and demand for public health nurses in Japan: A stock-flow approachPLOS ONE

Dear Dr. TAIRA,

Thank you for submitting your manuscript to PLOS ONE. After careful consideration, we feel that it has merit but does not fully meet PLOS ONE’s publication criteria as it currently stands. Therefore, we invite you to submit a revised version of the manuscript that addresses the points raised during the review process.

We look forward to receiving your revised manuscript.

Kind regards,

Botond Géza Kálmán, PhD

Academic Editor

PLOS ONE

Journal Requirements:

“This work was supported by FY 2021　Kusunoki 125 of Kyoto University 125th Anniversary Fund. The funders had no role in study design, data collection and analysis, decision to publish, or preparation of the manuscript.”

Reviewers' comments:

Reviewer's Responses to Questions

**Comments to the Author**

1. Is the manuscript technically sound, and do the data support the conclusions?

Reviewer #1: Yes

Reviewer #2: Partly

2. Has the statistical analysis been performed appropriately and rigorously?

Reviewer #1: I Don't Know

Reviewer #2: I Don't Know

3. Have the authors made all data underlying the findings in their manuscript fully available?

Reviewer #1: No

Reviewer #2: No

4. Is the manuscript presented in an intelligible fashion and written in standard English?

Reviewer #1: Yes

Reviewer #2: Yes

5. Review Comments to the Author

Reviewer #1: This is a well written, systematic study. Some questions however remain:

- In line 215, are these the confidence intervals? Or what is the ±?

- The text in table 1 is unclear

- Could the demand estimates not be made based on all of the independent variables (birth, cancer rate etc.?) instead of providing separate estimates for each?

- The response rate to the survey could be discussed in the methods section.

- line 215: you say "the number of PHN is 103", this seems rather low, could there be a clearer way to describe what you mean?

- line 26 and 27, talking in terms of change rate supply and demand is confusing for me. I would expect to hear something about the gap between total demand and supply.

Reviewer #2: This is an important topic and addresses crucial questions relevant to the Japanese population. However, the article may improved by addressing the following few issues:

1. Line 41, check and correct the mistakes in punctuation

2. Line 48 the statement is not clear. Consider paraphrasing it

3. Provide more details on the selection criteria of the survey sites in the methodology section.

4. In addition to the above, describe the characteristics of the personnel who were responding to the questionnaire at the site.

5. There is limited information provided on the contents of the questionnaire which was used in the study. The questionnaire should be included as supplementary file

6. In the results, number of PHN is presented in decimals. For example line 215 (103.2) and in table 1. May you explain this as it is confusing to the reader

7. Are the numbers presented in table 1 for the whole country? Provide further details to help readers interpret this table, preferably in as table legend

8. In line 255, include the n in blackets after the %

9. Line 269, this sentence is misplaced

10. In figure 1, I do not understand what NOT PHN and potential PHN means. Is NOT PHN everyone else who is not a PHN? Provide more information in a legend

There are more than three graphs in figure 1 and 2. Consider labelling them as 1A, 1B, 1C….

11. In addition to the above suggestion, think about it if it is necessary to have a separate graphs for estimates at 2025, 2030 and 2035. I am of the view that one graph can tell the whole story in a sequence

13. Provide title for figure 2 and figure 3

6. PLOS authors have the option to publish the peer review history of their article (what does this mean? ). If published, this will include your full peer review and any attached files.

**Do you want your identity to be public for this peer review?** For information about this choice, including consent withdrawal, please see our Privacy Policy .

Reviewer #1: No

Reviewer #2: No

---

## [Author Response · Author response to Decision Letter 0]

13 Jan 2024

Dear Editors and Reviewers,

Thank you for allowing us the opportunity to review our manuscript. Also, we are glad to have constructive and educational comments from you. We believe that our manuscript has been considerably improved thanks to your comments.

Below are our point-by-point responses to your comments and the manuscript with track change is uploaded as a supplementary file. Also, we found several other improvements in the wording in addition to the points suggested in the peer review, which we have also corrected. The corrections are summarized at the end of the document. Your comments have made the paper much better and more readable for the readers. We would appreciate it if you could check the revisions.

Response to Editor

and

Response: Yes. We checked that this paper follows the submission guidelines of PLOS ONE.

“This work was supported by FY 2021　Kusunoki 125 of Kyoto University 125th Anniversary Fund. The funders had no role in study design, data collection and analysis, decision to publish, or preparation of the manuscript.”

Response: We appreciate the information you have provided. In the cover letter, you declared that "There was no additional external funding received for this study."

Response: Thank you for pointing this out. We removed the following Ethical approval section.

【Removed: Page 23, Line 347 – 351】

Ethical approval and consent to participate

The ethical review of this study was waived by the Kyoto University Graduate School and Faculty of Medicine Kyoto University Hospital Ethics Committee because it was a questionnaire survey that did not involve personal information and secondary use of published data.

Response to Reviewers

Response for Reviewer #1:

#1_1. In line 215, are these the confidence intervals? Or what is the ±?

Response: Thank you for your question. Following convention, the description was written in the sense of mean ± standard deviation, but to avoid misunderstanding, we have modified these sentences to clearly expression.

【Original: Page 14, Line 214 – 218】

The number of full-time PHNs was 103.2 ± 75.9 in 2019 and 104.5 ± 76.4 in 2020. The number of part-time PHNs was 11.7 ± 14.9 in 2019 and 22.4 ± 53.6 in 2020. The number of PHNs who quit or retired was 5.0 ± 4.3 in 2019 and 4.9 ± 4.5 in 2020. The number of PHN recruitment examinations was 1.3 ± 1.0 in 2019 and 1.6 ± 1.1 in 2020 (Table 1).

【Revised: Page 15, Line 230 – 236】

The mean number of full-time PHNs for local governments was 103.2 (Standard Deviation: SD=75.9) in 2019 and 104.5 (SD=76.4) in 2020. The mean number of part-time PHNs for local government was 11.7 (SD=14.9) in 2019 and 22.4 (SD=53.6) in 2020. The mean number of PHNs who quit or retired for local government was 5.0 (SD=4.3) in 2019 and 4.9 (SD=4.5) in 2020. The number of PHN recruitment examinations for local government was 1.3 (SD=1.0) in 2019 and 1.6 (SD=1.1) in 2020 (Table 1).

#1_2. The text in table 1 is unclear

Response: We apologize for the broken layout of Talbe1 in the PDF proofing. After modifying the layout, the description of the mean and standard deviation has been modified to make it easier to understand as in comment #1_1. (Table 1)

#1_3. Could the demand estimates not be made based on all of the independent variables (birth, cancer rate etc.?) instead of providing separate estimates for each?

Response: Thank you for your insightful comments.　We also attempted to make predictions by multivariate analysis, but the small number of data used, and the confidence intervals were very large, and the analysis deemed inappropriate (Appendix). We believe that comparing models with Adjusted R2 values is also beneficial because it confirms that there are no major differences between univariate and multivariate analyses, and that looking at univariates does not change the trend when forecasts are made with different variables. However, since the construction of a more accurate forecasting model is an issue for the future, we have added it to the limitations section.

Appendix. The estimated number of public health nurses by multivariate analysis

Year The number of 95% Confidence Interval

public health nurses Lower Upper

2025 56227.8 -84201.7 226785.8

2030 56054.1 -87553.8 237146.9

2035 55982.7 -88812.2 243597.5

【Insert: Page 24, Line 347 – 349】

Fourthly, although simple univariate prediction models were used to estimate demand of PHNs in this study, more accurate model building might be done using multivariate analysis. However, more data needs to be accumulated, and this is an issue to be addressed in the future.

#1_4. The response rate to the survey could be discussed in the methods section.

Response: Thank you for your advice. We have left the response rate in the Results section because we believe it is an important part of the results in determining the adequacy of the underlying data used in the estimation. If there is a reason why they should be placed in the Methods section, we would appreciate it if you could point it out again.

#1_5. line 215: you say "the number of PHN is 103", this seems rather low, could there be a clearer way to describe what you mean?

Response: I appreciate your honest question. We have corrected the clarifying sentences because the number you pointed out is the mean number of PHNs working in the local government where the response was received. Based on the data in Figure 2 and S2, the number of PHNs in Japan in 2020 is 55,595. If the 150 local governments surveyed in this study employed an average of 103 public health nurses each, the total number of PHN would be 15,450. This figure is considered reasonable since there are approximately 1,500 small municipalities that do not have public health centers, which were excluded from the survey. The descriptions for inclusion and exclusion criteria of local governments added based on the reviewer comment #2_3.

【Insert: Page 7, Line 110–Page8, Line 118】

Local governments where public health centers are located consist of prefectures that unite cities, towns, and villages, and cities with large populations, such as ordinance-designated cities with populations of 500,000 or more and core cities with populations of 200,000 or more. Including other smaller cities, towns and villages, there are 1,718 local governments in Japan. However, only local government with public health centers were included in this survey because large local governments, which have more discretion over personnel costs, are more sensitive to increasing or decreasing the number of PHNs, and public health centers are also likely to be significantly affected by COVID-19 pandemic.

【Insert: Page 8, Line 122–124】

However, since the objective of this study was to determine the number of staff engaged in public health nursing services, we requested that PHNs and RNs working in hospitals and clinics operated by local governments be excluded from the count.

【Insert: Page 25, Line 364–368】

Thirdly, the survey for local governments conducted in this study was conducted for prefectures and large cities and did not take into account the number of PHNs in smaller local governments. However, smaller local governments were excluded from the study because some of them have no choice for increase or decrease in the number of PHNs, with only 1 or 2 PHNs assigned to them in the first place, and they may be affected by other factors, such as the shortage of human resources in rural areas.

#1_6 line 26 and 27, talking in terms of change rate supply and demand is confusing for me. I would expect to hear something about the gap between total demand and supply.

Response: I strongly agree with your suggestion. Fig. 3 has been partially revised to show the calculation process, and Fig. 4 has been

added to show the future demand and supply gap. Accordingly, the results of Fig. 4 have been added to the abstract and main text.

【Original: Abstract】

At the current job change rate, the supply of PHNs is expected to be 2,285 in 2025, 1,879 in 2030, and 1,908 in 2035. The demand for PHNs is estimated to be 2,335–2,779 in 2025, 2,314–2,886 in 2030, and 2,238–2,680 in 2035.

【Revised: Abstract】

At the current job change rate to PHNs, the gaps between future supply and demand of PHNs were expected to be -494 – -50 in 2025, -1,007 – -435 in 2030, and -772 – -330 in 2035. If the job change rate would be 1.5 times, the gaps were estimated to be 285 – 729 in 2025, -31 – 141 in 2030, and -182 – 260 in 2035. If the job change rate would be 2.0 times, the supply was sufficient for all estimated years.

【Insert: Page 21, Line 312–Page22, Line317】

Calculating the gap between supply and demand for future PHNs, the supply was insufficient for demand in all estimated years 2025, 2030, and 2035 when the current rate of job change to PHNs was used. If the job change rate to PHNs would be increased by 1.5 times, the supply was sufficient in 2025, but in 2030 and 2035, there was a shortage of up to 431 positions. Furthermore, if the job change rate would be increased to 2 times, the supply was sufficient for all estimated years (Fig 4).

Fig 4. Gaps of future supply and demand of public health nurses.

Response for Reviewer #2:

#2_1. Line 41, check and correct the mistakes in punctuation

Response: We apologize for the typographical error, which we have corrected.

【Original: Page 3, Line 41】

hospitals, most work in public health centers owned by the local government [2]. --PHNs in

【Revised: Page 3, Line 43】

hospitals, most work in public health centers owned by the local government [2]. PHNs in

#2_2. Line 48 the statement is not clear. Consider paraphrasing it

Response: Thank you for your point in improving readability. We have added the information because we believe it is necessary to explain the qualification system for public health nurses in Japan.

【Original: Page 3, Line 47–Page 4, Line 49】

Due to the nature of their duties and qualifications, PHNs also have difficulty employing foreigners like nurses and caregivers [6].

【Revised: Page 4, Line 49–53】

PHNs have difficulty hiring foreigners such as nurses and caregivers because other countries do not have PHN positions that perform the same duties as in Japan or the appropriate national qualifications, and they lack a system to help foreigners obtain PHN qualification [6].

#2_3. Provide more details on the selection criteria of the survey sites in the methodology section.

Response: Thank you very much for your critical suggestion. We added the sentences regarding the selection criteria for target local

government, PHNs and RNs to be counted, and the limitations of the study arising from the selection criteria.

【Insert: Page 7, Line 110–Page8, Line 118】

Local governments where public health centers are located consist of prefectures that unite cities, towns, and villages, and cities with large populations, such as ordinance-designated cities with populations of 500,000 or more and core cities with populations of 200,000 or more. Including other smaller cities, towns and villages, there are 1,718 local governments in Japan. However, only local government with public health centers were included in this survey because large local governments, which have more discretion over personnel costs, are more sensitive to increasing or decreasing the number of PHNs, and public health centers are also likely to be significantly affected by COVID-19 pandemic.

【Insert: Page 8, Line 122–124】

However, since the objective of this study was to determine the number of staff engaged in public health nursing services, we requested that PHNs and RNs working in hospitals and clinics operated by local governments be excluded from the count.

【Insert: Page 25, Line 364–368】

Thirdly, the survey for local governments conducted in this study was conducted for prefectures and large cities and did not take into account the number of PHNs in smaller local governments. However, smaller local governments were excluded from the study because some of them have no choice for increase or decrease in the number of PHNs, with only 1 or 2 PHNs assigned to them in the first place, and they may be affected by other factors, such as the shortage of human resources in rural areas.

#2_4. In addition to the above, describe the characteristics of the personnel who were responding to the questionnaire at the site.

Response: Thank you for your important suggestion. We added the sentences to describe the characteristics of personnel who were responding the questionnaire.

【Insert: Page 7, Line 102–106】

We identified from the target local governments' websites the department in charge of human resource management, such as recruitment, leave of absence, retirement, and staffing, and asked the department preparing the responses for questionnaire. In the Japanese government system, the responses were generally double-checked by the staff and manager in the department and the answer was given in the name of the head of the department.

#2_5. There is limited information provided on the contents of the questionnaire which was used in the study. The questionnaire should be included as supplementary file

Response: Yes, I have attached the questionnaire as Supplementary file 4 (S4). However, due to the large volume of the entire survey instrument, and in order to avoid leakage of research ideas and to be considerate of items that require permission for publication, we have excerpted only those items that are relevant to the analysis of this study.

【Original: Page 7, Line 101 – 104】

the survey items included the number of full-time PHNs hired, number of new graduates among the full-time PHNs hired, number of part-time PHNs hired, and number of full-time PHNs who quit or retired.

【Revised: Page 8, Line 119 – 122】

the survey items included the number of full-time PHNs hired, number of new graduates among the full-time PHNs hired, number of part-time PHNs hired, and number of full-time PHNs who quit or retired (S4).

【Insert: Page 33, Line 494 – 495】

S4. Survey on the Employment Status of Public Health Nurses (PHNs) during the Corona Disaster Survey Questionnaire

#2_6. In the results, number of PHN is presented in decimals. For example line 215 (103.2) and in table 1. May you explain this as it is confusing to the reader

Response: I appreciate your honest comments. We had received similar comments in Reviewer comment #1_5, so we revised the sentences to clearly convey our meaning.

【Original: Page 14, Line 214 – 218】

The response rate of the survey was 54.0% (81/150). For each variable, we compared the data for 2019 and 2020, that is, before and after the COVID-19 pandemic, respectively. The number of full-time PHNs was 103.2 ± 75.9 in 2019 and 104.5 ± 76.4 in 2020. The number of part-time PHNs was 11.7 ± 14.9 in 2019 and 22.4 ± 53.6 in 2020. The number of PHNs who quit or retired was 5.0 ± 4.3 in 2019 and 4.9 ± 4.5 in

---

## [Decision Letter · Decision Letter 1]

14 Nov 2024

PONE-D-23-19276R1Estimation of supply and demand for public health nurses in Japan: A stock-flow approachPLOS ONE

Dear Dr. TAIRA,

Thank you for submitting your manuscript to PLOS ONE. After careful consideration, we feel that it has merit but does not fully meet PLOS ONE’s publication criteria as it currently stands. Therefore, we invite you to submit a revised version of the manuscript that addresses the points raised during the review process.

Your revised manuscript has been assessed and the review comments are available below. Reviewer 3 has provided suggestions on  how to improve the clarity of your manuscript including the statistical models used. Please review their comments and make the appropriate revisions to your manuscript.

We look forward to receiving your revised manuscript.

Kind regards,

Emma Campbell, Ph.D

Staff Editor

PLOS ONE

Reviewers' comments:

Reviewer's Responses to Questions

**Comments to the Author**

1. If the authors have adequately addressed your comments raised in a previous round of review and you feel that this manuscript is now acceptable for publication, you may indicate that here to bypass the “Comments to the Author” section, enter your conflict of interest statement in the “Confidential to Editor” section, and submit your "Accept" recommendation.

Reviewer #2: All comments have been addressed

Reviewer #3: All comments have been addressed

Reviewer #4: (No Response)

2. Is the manuscript technically sound, and do the data support the conclusions?

Reviewer #2: Yes

Reviewer #3: Yes

Reviewer #4: Partly

3. Has the statistical analysis been performed appropriately and rigorously?

Reviewer #2: I Don't Know

Reviewer #3: Yes

Reviewer #4: No

4. Have the authors made all data underlying the findings in their manuscript fully available?

Reviewer #2: Yes

Reviewer #3: Yes

Reviewer #4: Yes

5. Is the manuscript presented in an intelligible fashion and written in standard English?

Reviewer #2: Yes

Reviewer #3: Yes

Reviewer #4: Yes

6. Review Comments to the Author

Reviewer #2: (No Response)

Reviewer #3: Thanks to the authors. I have no other comment.

This manuscript has no ethical problem. The ethical approval of the article was provided by the authors.

Reviewer #4: The manuscript could be further improved.

Line 89: The questionnaire(s) links/attachment(s) is/are to be cited.

Line 123, 130, 131-134: The weblink is to be provided.

Line 180-186: The equations could be fine-tuned to the following:

Demand Estimate at Ti=The number of additional public health nurses needed at Ti + The number of new public health nurses hired by local governments in 2020.

The number of additional public health nurses needed at Ti =The projected number of public health nurses needed at Ti − The number of public health nurses employed by local governments in 2020.

The projected number of public health nurses needed at Ti = [Regression coefficient × Demographic parameters at Ti] +Intercept.

Line 139 -147: The equations could be fine-tuned to the following:

Supply Estimate at Ti=The number of newly graduated public health nurses at Ti + The number of people transitioning to public health nursing roles at Ti.

The number of people transitioning to public health nursing roles at Ti = The potential public health nurse resource pool at Ti × Percentage of job changers entering public health nursing.

The human resource pool of potential public health nurses at Ti = The number of nurses within the employable age range at Ti−The number of nurses already working as public health nurses.

The numbering of Parameters 1-4 and Parameters 1-3 requires revision. If different, different number is to be given.

Line 159-161: The statement could be revised as ‘The age limit was set at 37 years, which is the average upper age limit for public health nurse recruitment examinations, calculated from the log data in recruitment information.’

Line 161: The website link is to be inserted.

Line 167-168: The statement could be improved as ‘The number of employed public health nurses within a five-year age group in 2020 was obtained from government statistics to estimate the number of nurses already working as public health nurses."

Line 209: The link is to be provided.

Line 213: ‘Them’ could be replaced with ‘total demands for 5 years’ where total demands for 5 years divided by 5

Table 1: IQR could be used in place of ‘minimum – maximum’.

Line 241: Dot is to be placed after close bracket for (SD=125.) Japan

Line 261: ?37 (unclear)

S1, S2, S3, S4 is to be named as S1_Fig, S2_Fig, S3_Fig, S4 _Fig

Line 272: The figure is to be cited

Figure 3: Data in 2025, 2030, and 2035 could be displayed in the figure.

7. PLOS authors have the option to publish the peer review history of their article (what does this mean? ). If published, this will include your full peer review and any attached files.

**Do you want your identity to be public for this peer review?** For information about this choice, including consent withdrawal, please see our Privacy Policy .

Reviewer #2: **Yes: ** Francis Kachidza Chiumia

Reviewer #3: No

Reviewer #4: No

---

## [Author Response · Author response to Decision Letter 1]

27 Nov 2024

Dear Editors and Reviewers,

Thank you for allowing us the opportunity to review our manuscript. Also, we are glad to have constructive and educational comments from you. We believe that our manuscript has been considerably improved thanks to your comments.

Below are our point-by-point responses to your comments and the manuscript with track change is uploaded as a supplementary file. Also, we found several other improvements in addition to the points suggested in the peer review, which we have also corrected. The corrections are summarized at the end of the document. Your comments have made the paper much better and more readable for the readers. We would appreciate it if you could check the revisions.

Response to Reviewer #4

#1. Line 89: The questionnaire(s) links/attachment(s) is/are to be cited.

#2. Line 123, 130, 131-134: The weblink is to be provided.

#7. Line 161: The website link is to be inserted.

#9. Line 209: The link is to be provided.

#15. Line 272: The figure is to be cited

Response: The reviewer comments (#1, 2, 7, 9 and 15) suggested adding links. While the submission guidelines for PLOS ONE do not explicitly mention this (URL), they instruct authors to use the Vancouver citation style. In this style, it is generally understood that URLs should be included in the reference list at the end of the manuscript. Accordingly, we have added citations with numbers where applicable but have not embedded URLs directly into the main text. If my understanding is incorrect, I would be happy to make the necessary adjustments, so please feel free to provide further instructions.

・Original: Page 6, Line 89–90

This was a simulation study based on a questionnaire survey of Japanese local governments, secondary data from government statistics, and log data from a website.

・Revised: Page 6, Line 89–90

This was a simulation study based on a questionnaire survey of Japanese local governments, secondary data from government statistics, and log data from a website [16–21].

・Original: Page 10, Line 159–Page 11, Line 161

The age limit was set at 37 years, which is the average upper age limit for the recruitment examinations for PHNs calculated from the log data from the recruitment information website.

・Revised: Page 11, Line 160–161

The age limit was set at 37 years, which is the average upper age limit for PHN　recruitment examinations, calculated from the log data in recruitment information [16].

・Original: Page 17, Line 270–274

The future number of PHNs needed, which is estimated based on the regression equation applied to each bivariate, is 58,567 in 2025, 60,251 in 2030, and 62,260 in 2035 based on the number of births; 13,397 in 2025, 16,904 in 2030, and 19,380 in 2035 based on the number of persons requiring certified long-term care; and 56,347 in 2025, 56,990 in 2030, and 57,257 in 2035 based on the estimated number of patients with cancer (S3, Fig 2).

・Revised: Page 11, Line 273–277

The future number of PHNs needed, which is estimated based on the regression equation applied to each bivariate, is 58,567 in 2025, 60,251 in 2030, and 62,260 in 2035 based on the number of births [20]; 13,397 in 2025, 16,904 in 2030, and 19,380 in 2035 based on the number of persons requiring certified long-term care [21]; and 56,347 in 2025, 56,990 in 2030, and 57,257 in 2035 based on the estimated number of patients with cancer [22] (S3_Fig, Fig 2).

ーーーーーーーーーーーーーーーーーーーーーーーーーーーーーーーーーーー

#3. Line 180-186: The equations could be fine-tuned to the following:

・Demand Estimate at Ti=The number of additional public health nurses needed at Ti + The number of new public health nurses hired by local governments in 2020.

・The number of additional public health nurses needed at Ti =The projected number of public health nurses needed at Ti − The number of public health nurses employed by local governments in 2020.

・The projected number of public health nurses needed at Ti = [Regression coefficient × Demographic parameters at Ti] +Intercept.

Response: Based on the fine-tuned equations you suggested, I have revised the formulas for demand calculations.

・Original: Page 12, Line 180–186

Demand Estimates Ti＝The number of increased PHNs demand at point Ti　+ The number of new PHNs hired by local governments in 2020

The number of increased PHNs demand at point Ti = The number of PHNs that would be needed at point Ti − The number of PHNs working in local government in 2020

The number of PHNs that would be needed at point Ti = Regression coefficient [B] × the demographic parameters Ti + [Intercept]

・Revised: Page 12, Line 183–189

Demand Estimates at Ti＝The number of additional public health nurses needed at Ti+ The number of new public health nurses hired by local governments in 2020

The number of additional public health nurses needed at Ti = The projected number of public health nurses needed at Ti − The number of public health nurses employed by local government in 2020

The projected number of public health nurses needed at Ti = [Regression coefficient × Demographic parameters at Ti] + Intercept

ーーーーーーーーーーーーーーーーーーーーーーーーーーーーーーーーーーーーーーーーーーーーーーーーーーーーーーーーーーーーーーーーーーーーーーーーーーーー

#4. Line 139 -147: The equations could be fine-tuned to the following:

・Supply Estimate at Ti=The number of newly graduated public health nurses at Ti + The number of people transitioning to public health nursing roles at Ti.

・The number of people transitioning to public health nursing roles at Ti = The potential public health nurse resource pool at Ti × Percentage of job changers entering public health nursing.

・The human resource pool of potential public health nurses at Ti = The number of nurses within the employable age range at Ti−The number of nurses already working as public health nurses.

Response: Based on the fine-tuned equations you suggested, I have revised the formulas for supply calculations.

・Original: Page 9, Line 139–Page 10, Line 147

Supply Estimates Ti ＝The number of newly graduated PHNs at point Ti + The number of people changing jobs to PHNs at point Ti

The number of people changing jobs to PHNs at point Ti=The human resource pool of potential PHNs at point Ti × The percentage of job changes to PHNs

The human resource pool of potential PHNs

=The number of PHNs within the age limit for employment at point Ti − The number of nurses already working as PHNs

・Revised: Page 9, Line 139–Page 10, Line 147

Supply Estimates at Ti ＝The number of newly graduated public health nurses at Ti + The number of people transitioning to public health nurses at Ti

The number of people transitioning to public health nurses at Ti =The potential public health nurse resource pool at Ti × Percentage of job changers entering public health nursing

The potential public health nurse resource pool at Ti =The number of nurses within the employable age range at Ti − The number of nurses already working as public health nurses

ーーーーーーーーーーーーーーーーーーーーーーーーーーーーーーーーーーー

#5. The numbering of Parameters 1-4 and Parameters 1-3 requires revision. If different, different number is to be given.

Response: Since the parameters related to supply (Parameter 1-4) and those related to demand (Parameter 1-3) were mixed, I have renamed them as 'Supply-related Parameter 1-4' and 'Demand-related Parameter 1-3.'

ーーーーーーーーーーーーーーーーーーーーーーーーーーーーーーーーーーー

#6. Line 159-161: The statement could be revised as ‘The age limit was set at 37 years, which is the average upper age limit for public health nurse recruitment examinations, calculated from the log data in recruitment information.’

Response: Thank you for your thoughtful suggestions on improving the English expressions. I agree with your suggestions and have revised the main text accordingly.

・Original: Page 10, Line 159–Page 11, Line 161

The age limit was set at 37 years, which is the average upper age limit for the recruitment examinations for PHNs calculated from the log data from the recruitment information website.

・Revised: Page 11, Line 160–161

The age limit was set at 37 years, which is the average upper age limit for PHN　recruitment examinations, calculated from the log data in recruitment information [16].

ーーーーーーーーーーーーーーーーーーーーーーーーーーーーーーーーーーー

#8. Line 167-168: The statement could be improved as ‘The number of employed public health nurses within a five-year age group in 2020 was obtained from government statistics to estimate the number of nurses already working as public health nurses."

Response: Thank you for your thoughtful suggestions on improving the English expressions. I agree with your suggestions and have revised the main text accordingly.

・Original: Page 11, Line 167–168

The number of employed PHNs within a five-year age group in 2020 was taken from government statistics to estimate the number of nurses already working as PHNs [19].

・Revised: Page 11, Line 169–161

The number of employed PHNs within a five-year age group in 2020 was obtained from government statistics to estimate the number of nurses already working as PHNs [19].

ーーーーーーーーーーーーーーーーーーーーーーーーーーーーーーーーーーー

#10. Line 213: ‘Them’ could be replaced with ‘total demands for 5 years’ where total demands for 5 years divided by 5

Response: Thank you very much for your suggestion. I determined that avoiding pronouns would improve readability, so I have revised the text as per your recommendation.

・Original: Page 14, Line 213–214

Since only five-year estimates could be calculated, we divided them by five and treated the average as the estimate for each year.

・Revised: Page 14, Line 216–217

Since only five-year estimates could be calculated, we divided the total demand for five years by five and treated the average as the estimate for each year.

ーーーーーーーーーーーーーーーーーーーーーーーーーーーーーーーーーーー

#11. Table 1: IQR could be used in place of ‘minimum – maximum’.

Response: Thank you for your insightful suggestion. I understand that it is common practice to report the IQR along with the median. However, in order to highlight the reality in certain local governments where the working environment was very challenging, I decided that including the maximum and minimum values would provide a clearer picture. Therefore, I have revised the table to include both the IQR and the maximum-minimum values.

ーーーーーーーーーーーーーーーーーーーーーーーーーーーーーーーーーーー

#12. Line 241: Dot is to be placed after close bracket for (SD=125.) Japan

Response: I apologize for the typo regarding the placement of the period. I have reviewed and corrected the indicated sentence as well as the entire text.

ーーーーーーーーーーーーーーーーーーーーーーーーーーーーーーーーーーー

#13. Line 261: ?37 (unclear)

Response: Thank you for your important feedback. Since the meaning of 'the number of PHNs under 37' was unclear, I have made the following revision.

・Original: Page 18, Line 259–262

For the number of people changing their job to PHNs, first, the number of PHNs under 37 is estimated to be 157,704 in 2025, 125,733 in 2030, and 116,752 in 2035.

・Revised: Page 18, Line 263–267

For the number of people changing their job to PHNs, first, the number of PHNs within the employable age range is estimated to be 157,704 in 2025, 125,733 in 2030, and 116,752 in 2035.

ーーーーーーーーーーーーーーーーーーーーーーーーーーーーーーーーーーー

#14. S1, S2, S3, S4 is to be named as S1_Fig, S2_Fig, S3_Fig, S4 _Fig

Response: To ensure consistency with the Supporting Information Captions, I have updated the references in the main text to S1_Fig, S2_Fig, S3_Fig, and S4_Fig.

ーーーーーーーーーーーーーーーーーーーーーーーーーーーーーーーーーーー

#16. Figure 3: Data in 2025, 2030, and 2035 could be displayed in the figure.

Response: Thank you for your feedback. I have added numerical values to the Figure 3.

ーーーーーーーーーーーーーーーーーーーーーーーーーーーーーーーーーーー

#Other revisions:

・The link for Reference 2 was broken, so it has been updated with a new one.

・An error was identified in the in-text citation numbering, where [29] was missing. This has been corrected.

・As a result of inserting the missing citation, the numbering for subsequent citations throughout the manuscript has been adjusted accordingly.

---

## [Decision Letter · Decision Letter 2]

5 Jan 2025

Estimation of supply and demand for public health nurses in Japan: A stock-flow approach

PONE-D-23-19276R2

Dear Dr. TAIRA,

We’re pleased to inform you that your manuscript has been judged scientifically suitable for publication and will be formally accepted for publication once it meets all outstanding technical requirements.

Kind regards,

Sergio A. Useche, Ph.D.

Academic Editor

PLOS ONE

Additional Editor Comments (optional):

Thanks for your amendments and clarifications. The paper can be accepted in its current form.

Reviewers' comments:

Reviewer's Responses to Questions

**Comments to the Author**

1. If the authors have adequately addressed your comments raised in a previous round of review and you feel that this manuscript is now acceptable for publication, you may indicate that here to bypass the “Comments to the Author” section, enter your conflict of interest statement in the “Confidential to Editor” section, and submit your "Accept" recommendation.

Reviewer #2: All comments have been addressed

Reviewer #3: All comments have been addressed

Reviewer #4: All comments have been addressed

2. Is the manuscript technically sound, and do the data support the conclusions?

Reviewer #2: Yes

Reviewer #3: Yes

Reviewer #4: Partly

3. Has the statistical analysis been performed appropriately and rigorously?

Reviewer #2: I Don't Know

Reviewer #3: Yes

Reviewer #4: Yes

4. Have the authors made all data underlying the findings in their manuscript fully available?

Reviewer #2: No

Reviewer #3: Yes

Reviewer #4: Yes

5. Is the manuscript presented in an intelligible fashion and written in standard English?

Reviewer #2: Yes

Reviewer #3: Yes

Reviewer #4: Yes

6. Review Comments to the Author

Reviewer #2: This is an interesting paper and it addresses an important topic with potential impact on policy. Thanks for your thorough address to the comments by all reviewers to this far. I have no further comments on the paper.

Reviewer #3: (No Response)

Reviewer #4: (No Response)

7. PLOS authors have the option to publish the peer review history of their article (what does this mean? ). If published, this will include your full peer review and any attached files.

**Do you want your identity to be public for this peer review?** For information about this choice, including consent withdrawal, please see our Privacy Policy .

Reviewer #2: No

Reviewer #3: **Yes: ** Nasrin Hanifi

Reviewer #4: No

---

## [Editor Report · Acceptance letter]

PONE-D-23-19276R2

PLOS ONE

Dear Dr. TAIRA,

I'm pleased to inform you that your manuscript has been deemed suitable for publication in PLOS ONE. Congratulations! Your manuscript is now being handed over to our production team.

Kind regards,

on behalf of

Dr. Sergio A. Useche

Academic Editor

PLOS ONE